Urban ecosystem quality assessment based on the improved remote sensing ecological index

Zhang Guolin
Kuang Honghai hhkuang@swu.edu.cn
School of Geographical Sciences, Southwest University , Chongqing , Asia , China
Brygadyrenko Viktor
Electronic publication date: 2025 Apr 29
Publication date: 2025
Volume: 13
Electronic Location ID: e19297
Received 2024 Dec 1; Accepted 2025 Mar 19
Copyright: ©2025 Zhang and Kuang
Copyright year: 2025
Copyright holder: Zhang and Kuang
License: This is an open access article distributed under the terms of the Creative Commons Attribution License, which permits unrestricted use, distribution, reproduction and adaptation in any medium and for any purpose provided that it is properly attributed. For attribution, the original author(s), title, publication source (PeerJ) and either DOI or URL of the article must be cited.
License URL: https://creativecommons.org/licenses/by/4.0/

Keywords: Remote sensing ecological index, Ecological index, Ecosystem quality, Google Earth engine, Chongqing city proper

Funding: The Southwest University F2022405 This work was supported by the Southwest University Grant F2022405. The funders had no role in study design, data collection and analysis, decision to publish, or preparation of the manuscript.

==============================
The remote sensing ecological index (RSEI) is an important tool for assessing ecosystem quality. However, its land surface temperature (LST) component poses challenges due to complex calculations and mismatched spatial resolution with other indicators. This study proposed an improved remote sensing ecological index (DRSEI). By replacing the LST component in RSEI with the difference index (DI) (representing PM2.5 concentration), the new index better reflects air pollution’s impact on ecosystem quality. The results demonstrated that DRSEI outperformed the RSEI in assessing ecosystem quality in Chongqing’s urban area. It exhibited three advantages: stronger correlation with the ecological index (EI), standard deviation values closer to EI’s baseline, and lower root mean square error. The applicability of the DRSEI and RSEI varied across different regions: the DRSEI proved to be more suitable for highly urbanized areas, whereas the RSEI performed better in suburban regions. Further analysis revealed that the spatial variability of indicators influenced their loadings in principal component analysis, thereby affecting ecosystem quality assessment results. This study emphasizes the importance of considering the spatial distribution of indicators when constructing ecological indices. The findings suggest DRSEI could effectively assess ecosystem quality in urbanized areas. This approach provides new insights for urban ecological monitoring and environmental management.

Introduction

The ecological environment serves as the foundation for human existence, and it directly impacts human health and socioeconomic development. Therefore, accurate assessment of the ecosystem quality (EQ) has become a crucial issue. The ecosystem quality refers to the ability of an ecosystem, within a specific time and spatial range, to maintain structural integrity and functional stability through the reasonable distribution and combination of its internal elements (PRC, 2015; PRC, 2021). It provides ecological services to safeguard human well–being while also possessing the ability to respond to and recover from external disturbances. The methods for evaluating EQ have evolved with the continuous development of data and technology. Many institutions and scholars worldwide have conducted extensive research in this field. Internationally, one of the most widely used models is the Pressure-State-Response (PSR) model, jointly proposed by the Organisation for Economic Co-operation and Development (OECD) and the United Nations Environment Programme (UNEP) (OECD, 1993; Tavosi et al., 2025). In China, the ecological index (EI) was introduced in 2015 (PRC, 2015), followed by the trial implementation of the ecosystem quality index (EQI) in 2021 (PRC, 2021). The PSR model requires indicators that cover multiple dimensions, including pressure, state, and response. However, the acquisition of these indicators often faces challenges such as data scarcity and difficulty in collection. Similarly, both EI and EQI encounter issues related to data acquisition, complex indicator construction, and spatial scale limitations. To address these issues, Xu (2013a) introduced the remote sensing ecological index (RSEI), which integrates four indicators: greenness, humidity, dryness, and heat. Unlike the EI and the EQI, the RSEI is a fully remote sensing–based composite index with easily obtainable indicators, without requiring manual weight assignment, and visualizable results. The RSEI’s applications span diverse ecosystems: grasslands (Du et al., 2024; Song, Luo & Duan, 2019), wetlands (Jing et al., 2020; Qureshi et al., 2020), urban areas (Huang et al., 2021; Xu, 2013b), arid regions (Gao et al., 2020; Xia et al., 2022), basins (Wang & Ge, 2022; Yuan et al., 2021), and mining areas (Nie et al., 2021; Zhu et al., 2020).

Although the RSEI model has been widely applied since its introduction, it has not been without flaws (Zhong & Xu, 2021). Since its inception, the model has attracted significant attention from researchers, who have proposed various improvements. However, the effectiveness and rationality of these improvements have been a topic of controversy in different studies. For example, some research attempted to enhance the RSEI by increasing the number of principal components (Song, Luo & Duan, 2019). However, Xu, the original author of the RSEI model, demonstrated through experiments that increasing the number of principal components not only reduced the proportion of the first principal component but also failed to increase the information content of the original RSEI. On the contrary, it led to interference between the components, thus not significantly improving the model’s performance. Modifying the combination of ecological factors is currently one of the most common ways to improve the model (Xu & Deng, 2022). Researchers added other indicators to the RSEI framework based on different research objectives. For instance, Wan et al. (2021) argued that particulate matter in the air influences the urban ecological environment, added the difference index (DI), representing PM2.5 concentration, to the RSEI to construct the RSEINew for urban ecological environment research. The results showed that the average correlation of the RSEINew with other indicators was higher than the RSEI. However, the authors’ validation of RSEINew focused solely on principal component analysis. While they examined contribution rates and factor loadings, they omitted critical comparisons between RSEINew/RSEI and established indices like EI or EQI. This validation was neither sufficient nor comprehensive, leaving uncertainty regarding whether the RSEINew has higher evaluation accuracy than the RSEI. Zhang et al. (2023) noted the lack of quantitative research combining air pollutant indicators with ecological environment status in EQ monitoring. Therefore, the aerosol optical depth (AOD) was incorporated into the RSEI, resulting in the improved remote sensing ecological index (ARSEI) to study the ecosystem quality of Xi’an. However, the authors only justified the rationality of the ARSEI by comparing the differences between the ARSEI and the RSEI, without verifying whether the ARSEI had higher evaluation accuracy than the RSEI. Similarly, Wang et al. (2022), recognizing the growing air pollution issue, incorporated the air pollution index (API) into the RSEI to develop the AQRSEI. By comparing the spatiotemporal visualization outputs of AQRSEI and RSEI models within ecological restoration regions, the author demonstrated that AQRSEI exhibits enhanced sensitivity in detecting spatiotemporal ecological variations. However, Wang et al. (2022) also did not compare the AQRSEI with the EI or other standard indices, lacking a quantitative metric to assess the improvements of the AQRSEI over the RSEI. While these studies have made certain improvements to the RSEI model, they generally lack comprehensive validation of the improvement effects.

Some scholars have incorporated indicators that highlight regional characteristics based on the specific features of different study areas. For example, the rocky desertification index (RI) (Ye & Kuang, 2022), Salinity (SI–T) and land degradation indicators (Wang et al., 2020), the net primary productivity of vegetation (NPP) (Fan et al., 2021), the human activity intensity index (IPOI) (Yang et al., 2021), and population density data (POP) (Zhao et al., 2022). This improvement aims to more comprehensively reflect the complexity of the ecological environment. However, these methods are usually applied only to specific regions. For example, the RI index is primarily used in the karst areas of southeastern Chongqing, while the SI-T and land degradation indicators are mainly applied in the Ulan Buh Desert. Similarly, studies that have modified the RSEI by incorporating NPP, IPOI, and POP have also been limited to a single study area. However, ecological issues such as rocky desertification and desertification are not confined to a single region; they occur across different latitudes, climatic conditions, and levels of human disturbance. Due to variations in natural and human environments among different regions, most existing studies on RSEI improvement methods lack systematic validation of their applicability across diverse geographical settings. This limitation restricts the generalization and broader applicability of these improved methods.

Building on the improvement methods proposed by various scholars, this study aims to refine the RSEI by adjusting its indicator composition. The original RSEI aligns with EI’s framework through three key correspondences: (1) NDVI as greenness (matching EI’s vegetation coverage), (2) WET index (equivalent to EI’s hydrographic network), and (3) NDSI representing dryness (analogous to EI’s land degradation indicator). Notably, EI’s pollution load index lacks a counterpart in the RSEI framework. The RSEI originally adopted land surface temperature (LST) as the heat indicator because its developers argued that, although heat indicators had not received sufficient attention in China’s ecological monitoring and were not included in environmental statistical yearbooks, thermal pollution (e.g., the urban heat island effect) remains an important factor (Xu, 2013a). However, with the continuous advancement of urbanization, air pollution control has become a core priority in China’s “ecological civilization” initiatives (Feng et al., 2019). Furthermore, the 2015 revision of the EI assigned the highest weight to air quality compliance in the urban ecological status index, highlighting the significant role of air pollution in urban ecological quality assessments. Therefore, when evaluating urban ecological quality, the impact of air pollution should be considered. To reduce model complexity, we replaced the heat indicator with an air pollution metric in the RSEI framework. This substitution aims to better capture air pollution’s ecological impacts without introducing additional variables.

The selection of air quality indicators also requires careful consideration. The most commonly used remote sensing product representing air quality is the aerosol optical depth (AOD) (Feng, Yang & Li, 2023; Liu et al., 2022; Zhang et al., 2023). However, Xu (2008) highlighted a limitation: coarse-resolution products inadequately capture urban aerosol dynamics. Due to spatial variations in factors such as buildings, transportation infrastructure, and population density, the aerosol distribution may vary at even finer spatial scales within urban areas (Xu et al., 2018). Although Sentinel–2 Satellite data provide higher spatial resolution AOD products than other satellites, their time series has been too short for long-term AOD studies (Li et al., 2019). Similarly, while Sentinel-5P’s atmospheric pollution index (API) product includes four gaseous pollutants (NO2, SO2, O3, and CO), it also suffers from coarse resolution and a short time series (Wang & Ge, 2022). In comparison, the difference index (DI) offers distinct advantages. It effectively captures PM2.5 concentrations while maintaining spatial resolution compatibility with other indicators. Furthermore, its computational simplicity (Feng, Feng & Feng, 2018) makes it particularly suitable for integrated analysis. This may provide a breakthrough for improving the RSEI. Therefore, we developed the DI to represent air quality parameters and constructed a novel remote sensing ecological index, the DRSEI. Given that PM2.5 pollution dominates Chongqing’s air pollution, and that high PM2.5 concentration areas are concentrated in the central urban district (Chen et al., 2022; Xiong et al., 2022), this study selected Chongqing’s main urban area as the research area. The objective was to explore the DRSEI’s effectiveness in improving urban EQ assessment and to evaluate the EQ of Chongqing’s central urban district. The findings of this study could support the urban ecological management. Portions of this text were previously published as part of a preprint (https://doi.org/10.21203/rs.3.rs-4756211/v1).

Materials & Methods

Study area

The municipal area in Chongqing covers an area of 82,400 km2 and is divided into three main functional zones: the city proper, the Three Gorges Reservoir area in the northeast, and the Wuling Mountain area in the southeast. Chongqing’s city proper comprises 21 districts, including the central urban area (Yuzhong, Dadukou, Jiangbei, Nan’an, Shapingba, Jiulongpo, Beibei, Yubei, and Ba’nan) and the main urban area (Fuling, Changshou, Jiangjin, Hechuan, Yongchuan, Nanchuan, Qijiang, Dazu, Bishan, Tongliang, Tongnan, and Rongchang), covering a total area of 28,700 km2 (Fig. 1). Accounting for 35% of Chongqing’s land area, this region accommodates 66% of the city’s permanent residents and 74% of the city’s urban population, and it contributes 77% of the regional GDP. It is characterized by a relatively high level of urbanization and is closely linked to urban development.

Figure 1 Location of the city proper in Chongqing.

Data and preprocessing

In this study, data for the city proper in Chongqing were collected for the years 2002, 2006, 2010, 2014, 2018, and 2022. Table 1 summarizes the key data used in this study (land cover data and statistical data for calculating the EI values). Due to the similarity of the annual trends of the RSEI and the average summer RSEI values, the data for June to August in each target year were selected (Ji et al., 2022; Zhang et al., 2021).

Table 1 Detailed description of the data.

Data types	Source	
Vector map data	Alibaba Cloud Visualization Platform
(https://web.archive.org/web/20250409042050/https://datav.aliyun.com/portal/school/atlas/area_selector, accessed April 2)	
LC05	Google Earth Engine (https://developers.google.com/earth-engine/datasets/catalog/LANDSAT_LT05_C02_T1_L2, accessed April 16)	
LC08	Google Earth Engine (https://developers.google.com/earth-engine/datasets/catalog/LANDSAT_LC08_C02_T1_L2, accessed April 19)	
Land cover data	The China Land Cover Dataset
(http://doi.org/10.5281/zenodo.4417809, accessed May 8)	
Statistical data	The Chongqing Water Resources Bureau
(https://data.stats.gov.cn/english/, accessed June 3)	
The Chongqing ecological environment bureau
(https://data.stats.gov.cn/english/, accessed June 17)	
The Chongqing statistics bureau
(https://data.stats.gov.cn/english/, accessed June 29)	

All image processing was performed on Google Earth Engine (GEE) using Landsat 5 (LC05) and Landsat 8 (LC08) data. The specific dataset identifiers were: LANDSAT/LT05/C02/T1_L2 for Landsat 5 (EROS, 2020a), and LANDSAT/LC08/C02/T1_L2 for Landsat 8 (EROS, 2020b). Prior to release, both satellite datasets underwent radiometric calibration and atmospheric correction. These preprocessing steps effectively eliminated sensor response variations, temporal imaging differences, and atmospheric interference, ensuring accurate representation of surface reflectance. LC05 and LC08 images were calibrated from sensor radiance to surface reflectance using the Landsat Ecosystem Disturbance Adaptive Processing System (LEDAPS) (Masek et al., 2006) and the Landsat Surface Reflectance Code (LaSRC) (Vörösmarty, Pahl-Wostl & Bhaduri, 2013), respectively. These datasets underwent preprocessing steps.

Due to Chongqing’s humid and rainy climate, the images were often affected by clouds. To address cloud contamination, we implemented the Fmask algorithm for cloud removal (Foga et al., 2017; Mateo-García et al., 2018; Zhu & Woodcock, 2012). In GEE, Fmask is encapsulated as a convenient function, allowing cloud and cloud shadow masking by extracting the “StateQA” band. After cloud removal, the images were composited using the median value. The effectiveness of cloud removal was evaluated through cloud cover calculations and visual analysis. Specifically, the “QA_PIXEL” band was used to identify cloudy pixels, and cloud coverage was calculated as the ratio of cloudy pixels to the total number of pixels. The results showed that after cloud removal, the cloud coverage of the six composite images from 2002 to 2022 was close to zero, meeting the requirements for further analysis. Since the cloud removal method strictly filters out cloud-contaminated areas, regions that are severely affected by cloud contamination in all available images may lose valid data after cloud removal, resulting in missing areas. Therefore, after cloud removal, it is necessary to fill the missing areas using images from the same month of the previous or following year.

Calculation of various indicators

(1) NDVI: The NDVI has been widely used to assess and monitor the health and coverage of surface vegetation. It is calculated as follows: (1) NDVI=BNir−BRedBNir+BRed

where BNir is the reflectance in the near–infrared band, and BRed is the reflectance in the red band.

(2) WET: The moisture index not only represents open water bodies but is also closely related to the moisture in the soil and vegetation. Before extracting the WET index, water bodies are masked using the modified normalized difference water index (Xiong et al., 2021) (MNDWI, Eq. (3)). The formula for calculating the WET index (Crist, 1985) is WETTM=0.0315BBlue+0.2021BGreen+0.3102BRed

+0.1594BNir−0.6806BSwir1−0.6109BSwir2

WETOLI=0.1511BBlue+0.1973BGreen+0.3283BRed

(2) +0.3407BNir−0.7117BSwir1−0.4559BSwir2

(3) MNDWI=BGreen−BSwir1BGreen+BSwir1

where BBlue, BGreen, BRed, BNir,  BSwir1, and BSwir2 are the reflectances of the Landsat data in the blue, green, red, near–infrared, shortwave infrared 1, and shortwave infrared 2 bands, respectively.

(3) LST: The LST represents the surface temperature in an area. In this study, the Landsat LST product (Ermida et al., 2020) provided by the GEE platform was utilized, requiring only conversion to Celsius. The calculation formula is as follows: (4) LST=BST−273.15

where BST is the land surface temperature band (for the LANDSAT/LT05/C02/T1_L2 dataset, it is the ST_B6 band, and for the LANDSAT/LC08/C02/T1_L2 dataset, it is the ST_B10 band). To convert the temperature to Celsius, 273.15 is subtracted from the Kelvin value.

(4) NDSI: The NDSI consists of two components: the scarcity index (SI), which reflects the extent of vegetated land scarcity (Rikimaru, Roy & Miyatake, 2002); and the index-based built-up index (IBI), which reflects built-up land conditions (Xu, 2008). Its calculation formula is

NDSI=SI+IBI2

SI=BSwir1+BRed−BBlue+BNirBSwir1+BRed+BBlue+BNir

(5) IBI=2BSwir1BSwir1+BNir−BNirBRed+BNir+BGreenBSwir1+BGreen2BSwir1BSwir1+BNir+BNirBRed+BNir+BGreenBSwir1+BGreen

where BBlue, BGreen, BRed, BNir, BSwir1, and BSwir2 are the reflectances of the Landsat data in the blue, green, red, near–infrared, shortwave infrared 1, and shortwave infrared 2 bands, respectively.

(5) DI: Feng, Feng & Feng (2018) developed the particle difference index (DI) to represent changes in particle concentration based on the characteristics of the PM2.5, which increases the reflectance in the red band and decreases the reflectance in the near–infrared band. Its calculation formula is (6) DI=BRed−BNir

where BRed and BNir are the apparent reflectance or radiance in the red and near–infrared bands, respectively.

DRSEI comprehensive index development

The four indicators included in the RSEI are greenness, moisture, dryness, and heat. Greenness is represented by the normalized difference vegetation index (NDVI), moisture is represented by the humidity index obtained through remote sensing tasseled cap transformation (WET), dryness is represented by the average of the soil index (SI) and the built–up index (IBI) (NDSI), and heat is represented by the land surface temperature (LST) derived from Landsat. Thus, the definition of the RSEI is (Xu, 2013a). RSEI=fNDVI,WET,LST,NDSI.

By replacing LST in the RSEI with the difference index (DI), a new type of remote sensing ecological index is constructed, denoted as the DRSEI. Therefore, the definition of the DRSEI is DRSEI=fNDVI,WET,NDSI,DI.

To integrate these four indicators into a single index, the principal component analysis (PCA) method (Bro & Smilde, 2014) is commonly employed. PCA identifies the correlations among these indicators by analyzing their variations and determines their respective weights based on each indicator’s contribution to the principal components. This approach transforms the original data into a few representative principal components, thereby reducing data complexity and better capturing the overall ecological environment. Therefore, the formula for the DRSEI is: (7) DRSEI=PC1fNDVI,WET,NDSI,DI

where PC1 represents the first principal component obtained from PCA.

For the calculated four indicators, due to their non–uniform dimensions, it was necessary to normalize these indicators before PCA (Ye & Kuang, 2022). That is, their values needed to be normalized to the range of [0,1]. The normalization formula is (8) Ni=Ii−IminImax−Imin

where Ni is the normalized value of a certain indicator, Ii is the value of that indicator in pixel i, and Imin and Imax are the minimum and maximum values of that indicator, respectively.

PCA was performed on the normalized indicators to retain the primary information in the data while achieving reduction of the dimensionality. To enable measurement and classification of the DRSEI, it was also normalized. The formula for this normalization is (9) DRSEI=DRSEIi−DRSEIminDRSEImax−DRSEImin

where DRSEIi is the value of DRSEI for pixel i, and DRSEImin and DRSEImax are the minimum and maximum values of DRSEI, respectively.

Model verification

The optimization effect of DRSEI, as an improved version of RSEI, can be validated from two dimensions:

1. Accuracy improvement assessment: using the ecological index (EI) as the evaluation standard, the improvement of DRSEI in ecological quality assessment is quantitatively evaluated by comparing the degree of convergence between DRSEI, RSEI, and EI. The selection of EI as the reference benchmark is based on its widespread adoption in China’s environmental statistical yearbooks for ecological quality assessment. The fit between DRSEI, RSEI, and EI is examined using three core indicators, with specific calculation formulas provided in Table 2: correlation coefficient, root–mean–square difference, and standard deviation.

Table 2 The calculation formulas for the evaluation indicators.

Index	Formula	Description	
Correlation coefficient	R=1N∑n−1Nfn−f¯pn−p¯δfδp	fn and pn are defined as scattered points at N times or as spatial points. f¯ and p¯ are the mean values of f and p, and δf and δp are the standard deviations of f and p, respectively.	
Average correlation	C¯p=Cq+Cr+⋯+Csn−1	C¯p is the average correlation coefficient. p, q, r, and s represent the average correlation coefficients and are the indicators used in the correlation analysis. Cq, Cr, andCs are the correlation coefficients between each pair of indicators.	
Standard deviation	SD=1N∑i=1Nxi−x¯2	SD is the standard deviation, N is the total number of data points, xi is the value of the ith data point, and x¯ is the mean of the dataset.	
Root–mean–square (RMS) difference	E′=1N∑n=1Nfn−f¯−pn−p¯22	The Taylor diagram suggests that the traditional root mean square error (RMSE) is composed of two components, namely, the overall bias (E¯) and the root–mean–square (RMS) difference (E′). N,fn,pn,f¯, and p¯ have the same meanings as in the correlation coefficient formula.	

2. Regional applicability analysis: a multi-regional comparative experiment is conducted to analyze the applicability differences between DRSEI and RSEI in regions with distinct ecological characteristics, assessing their stability and generalizability in complex geographical environments. Specifically, the DRSEI and RSEI scores for different ecological regions are calculated and systematically compared with the EI reference values for these regions to reveal the adaptability of the models under varying geographical conditions.

Results

Comparison of DRSEI and RSEI in the city proper in Chongqing

PCA was implemented on the GEE platform (Google Earth Engine) to obtain the loading coefficients of each indicator and the contribution of the first principal component (Table 3). The results indicate that the first principal component of the DRSEI accounted for over 75% of the total variance, demonstrating a capacity equivalent to that of the RSEI in synthesizing multi-indicator information. Compared to the RSEI, the higher variance contribution of the first principal component in the DRSEI suggested that its model accounted for a larger proportion of the total variance. This implied that the DRSEI summarized the data’s main characteristics more effectively than the RSEI. Consequently, it enhanced the capability to characterize ecological factor trends. Regarding loading polarities, greenness and wetness showed positive loadings, indicating beneficial contributions to the ecosystem, while dryness and air pollution indices exhibited negative loadings, reflecting detrimental impacts on EQ. These findings aligned closely with the RSEI pattern. Additionally, across all study years, the consistent signs of indicator loadings in the DRSEI suggested that the model had a stable capacity to reflect EQ evolution.

Table 3 PCA results for RSEI and DRSEI.

Year	Model	Contribution rate of PC1	Loadings of each indicator on PC1	
			NDVI	WET	NDSI	LST	DI	
2002	RSEI	76.52	0.80	0.13	–0.50	–0.30		
DRSEI	79.44	0.74	0.08	–0.44		–0.50	
2006	RSEI	74.54	0.79	0.11	–0.57	–0.20		
DRSEI	79.70	0.80	0.11	–0.57		–0.16	
2010	RSEI	84.69	0.81	0.07	–0.55	–0.18		
DRSEI	85.33	0.79	0.06	–0.52		–0.32	
2014	RSEI	81.54	0.74	0.13	–0.54	–0.37		
DRSEI	83.23	0.72	0.11	–0.50		–0.47	
2018	RSEI	85.85	0.93	0.11	–0.28	–0.20		
DRSEI	85.89	0.75	0.08	–0.22		–0.62	
2022	RSEI	90.33	0.94	0.13	–0.16	–0.28		
DRSEI	86.16	0.76	0.09	–0.13		–0.63	

Both the contribution rate of the first principal component and PCA-derived indicator loadings suggested that the DRSEI enhanced multi-indicator integration; however, its validity for EQ evaluation required further verification. We selected 2018 and 2022 as representative cases where both indices showed high contribution rates, then conducted comparative analyses between the indices and their corresponding EI values across administrative divisions.

At the county level, we computed correlation coefficients, RMS differences, and standard deviations between EI and both RSEI/DRSEI for 2018 and 2022. The correlation coefficient revealed which model better captured the authentic ecological variation trends overall. RMS differences quantified deviations between EI and the indices (EI-RSEI/DSEI pairs), while standard deviations measured the magnitude of data dispersion, thereby reflecting the spatial heterogeneity of ecological indices. To more intuitively illustrate the relationships among the three evaluation indices, we used a Taylor diagram (Fig. 2). The Taylor diagram can simultaneously display the correlation, standard deviation, and RMS difference of multiple models relative to the reference value (EI) within the same coordinate system, overcoming the limitations of traditional tables or single scatter plots in comprehensively comparing multiple indices. Each point in the diagram represents a model. The angle between a radial axis and the abscissa represents the correlation coefficient. Smaller angles (approaching 0°) indicate values closer to 1, denoting stronger linear consistency with ground truth data. The distances along orthogonal axes denote standard deviations. Euclidean distances between model markers and the reference point represent RMS differences; shorter distances indicate smaller RMS errors (Taylor, 2001).

Figure 2 Comparison of accuracy between RSEI and DRSEI.

First, we compared the standard deviations in the Taylor diagram. Analysis of the relative positions between scatter points and the solid arc connected to the REF point in Fig. 2 revealed that standard deviations followed the order: RSEI >DRSEI >EI. However, the differences among them were relatively small, indicating that while both DRSEI and RSEI exhibited slightly greater data variability than EI, the extent of this difference was limited. This suggests that these models may slightly amplify the magnitude of ecological changes. Nevertheless, the comparison demonstrates the DRSEI’s standard deviation is closer to EI’s value, suggesting its superior realism in EQ representation. Next, we compared the correlation coefficients, which correspond to the angles of the radial lines in the diagram. The results showed that both indices had correlation coefficients of approximately 0.6 with the EI, indicating moderate correlations. Notably, the DRSEI exhibited a higher correlation coefficient than the RSEI, which suggests that the spatial distribution of the DRSEI was more synchronized with the actual EQ represented by the EI. Finally, we examined the root-mean-square (RMS) difference, which quantifies the overall discrepancy between the model values and the actual measurements. This metric is visualized in the diagram as the distance between scatter points and the reference (REF) point. Comparison of scatter point positions relative to the pink arc shows the DRSEI exhibiting smaller RMS difference, demonstrating superior fitting accuracy and spatial consistency in EQ assessment. These findings further validated the improvements in the DRSEI.

Compared to the RSEI, the DRSEI exhibited two key improvements in EQ assessment: fluctuations closer to true values; and higher spatial synchronization coupled with smaller overall error. These findings demonstrated its improved accuracy.

Comparison of DRSEI and RSEI in other cities

The DRSEI performed well in the main urban area of Chongqing, but its effectiveness might vary across cities with distinct geographical locations and climatic conditions. To systematically compare the DRSEI and the RSEI, this study selected Beijing (a northern city) and Guangzhou (a southern city) for comparison. Beijing is located in the northern North China Plain, while Guangzhou lies in the coastal area of South China. These significant latitudinal and geographical contrasts provided an ideal foundation for assessing the applicability of the indices across diverse environments. For Beijing, district-level ecological index (EI) data for 2022 were obtained from the Beijing Ecological Environment Status Bulletin. In contrast, Guangzhou’s EI reporting ceased after 2020, requiring the use of 2020 data. Differences between DRSEI, RSEI and EI were calculated and visualized using lollipop plots, explicitly demonstrating their discrepancies, as shown in Fig. 3. The lollipop plot not only intuitively displays the magnitude of the deviations of DRSEI and RSEI from EI across districts but also clearly indicates the direction of these deviations.

Figure 3 Differences between DRSEI and EI vs. RSEI and EI.

Figure 3 reveals that both DRSEI and RSEI show limited effectiveness in evaluating Beijing’s district-level EQ. Both models demonstrate systematic underestimation across the municipality. However, three key observations emerge: First, discrepancies exist between citywide and district-level evaluations. For both models, citywide EQ assessments exhibited higher accurate compared to district-level evaluations. Second, at the district level, the DRSEI achieved higher accuracy in central urban areas (including Dongcheng, Xicheng, Chaoyang, Fengtai, Shijingshan, Haidian, Mentougou, and Daxing). Smaller discrepancies between DRSEI and EI values in these districts demonstrate its enhanced suitability for urbanization-dominated ecosystems. In contrast, the RSEI demonstrated higher accuracy in assessing Beijing’s suburban districts (Fangshan, Shunyi, Huairou, Pinggu, Miyun, and Yanqing). These areas typically feature higher vegetation cover and lower urbanization levels, suggesting that the RSEI may be more suitable for reflecting EQ in such natural environments.

However, Guangzhou’s EQ assessment revealed marked disparities between DRSEI and RSEI performances. The RSEI generally overestimated EQ, while the DRSEI consistently underestimated it. This systematic divergence may stem from differences in model sensitivity to ecological factors under the humid and vegetation-rich climatic of southern China. Compared to district-level evaluations, the DRSEI demonstrated greater accuracy in citywide EQ assessments. This again demonstrates that, at a large scale, the DRSEI model can more stably capture the overall trend of ecological change. Additionally, the DRSEI showed significantly higher accuracy than the RSEI in assessing Guangzhou’s central urban districts, including Liwan, Yuexiu, Haizhu, Tianhe, and Baiyun. This spatial consistency with Beijing’s pattern strengthens evidence for differential model applicability across urbanization gradients. The DRSEI is better suited for assessing EQ in highly urbanized areas, while the RSEI performs better in natural ecological environments.

As DRSEI and RSEI demonstrate distinct regional applicability advantages and diverge in key metrics (DI for DRSEI versus LST for RSEI), systematically investigating LST and DI as potential core drivers becomes critical for elucidating the mechanistic basis of model performance variations. To validate LST and DI impacts on EQ, we first analyzed correlations among these parameters and EI (Fig. 4). The correlation analysis aims to assess whether LST and DI exhibit equivalent significance for ecological quality, thereby investigating their roles in the performance discrepancies between models. Should DI demonstrate a statistically stronger correlation with EI compared to LST, this would imply that the superiority of DRSEI in ecological quality assessment may originate from the validity of DI as a surrogate indicator. Specifically, DI could provide more precise characterization of ecological conditions, whereas LST might play a comparatively subordinate role. Conversely, if LST and DI show comparable correlations with EI, it would suggest both parameters exert similar influences on ecological quality. In such cases, the accuracy differences between DRSEI and RSEI might not be exclusively attributed to the substitution of evaluation factors, but could also be attributable to additional confounding variables.

Figure 4 Correlation comparison: DI and EI vs. LST and EI.

Beijing showed DI-EI and LST-EI correlation coefficients of −0.92 (R2 = 0.86) and −0.93 (R2 = 0.87), respectively. Guangzhou demonstrated stronger DI-EI (−0.96, R2 = 0.93) and LST-EI (−0.96, R2 = 0.92) correlations. The consistently strong negative correlations (high R2 values) between DI/LST and EI across both cities confirm their significant influence on EQ. These robust correlations substantiate the rationale and feasibility of substituting the DI for the LST in EQ assessments. Notably, despite the strong correlations between the DI and the LST with the EI in both cities, the overall evaluation results of the DRSEI and the RSEI in Guangzhou showed substantial differences. This suggested that the discrepancy in evaluation accuracy was not entirely caused by the fundamental relationship between the LST and the DI with EQ.

The PCA weighting mechanism is influenced by the spatial variability of indicators: regions with greater variation exhibit higher variance contributions and loading weights. This necessitates examining how indicators’ spatial distribution characteristics interact with the model’s weighting mechanism (Zheng et al., 2022). From a geospatial perspective, this study aimed to analyze the formation path of model differences. To elucidate this mechanism comprehensively, the study integrates analysis of spatial frequency distributions and principal component loadings for both DRSEI and RSEI across Guangzhou and Beijing. The spatial frequency histograms (Fig. 5) characterize the value distribution patterns of individual indicators across distinct geographical zones, particularly revealing whether specific indicators demonstrate localized concentration or extensive dispersion. The principal component loadings (Table 4), conversely, quantify each indicator’s contribution during the PCA weighting procedure. By synthesizing these spatial distribution patterns with their corresponding statistical weights, this investigation systematically demonstrates how spatial heterogeneity of indicators modulates PCA weighting schemes, and further reveals how such modulations generate regional discrepancies in model applicability between DRSEI and RSEI. These findings collectively advance our understanding of the mechanistic differences underlying the two indices.

Figure 5 Frequency distribution histograms of individual indicators.

The vertical axis represents the percentage of frequency, and the horizontal axis represents the values.

Since the NDVI, WET, and NDSI are included in both models, this study focused on the differences between the LST and the DI. As shown in Fig. 5, LST and DI exhibit significant spatial variations in Beijing; their data distribution patterns show notable similarity. When combined with the overall scores of the DRSEI and the RSEI in Beijing shown in Fig. 3, the two models produce very similar scores. In contrast, in Guangzhou, the LST shows minimal variation, while the DI exhibits noticeable fluctuations. The higher evaluation accuracy of DRSEI in Guangzhou (Fig. 3) suggests that DI differences may critically influence model performance. Notably, LST and DI spatial distributions correlate with model evaluation results: DRSEI and RSEI scores converge when their distributions align, whereas distribution divergences amplify score discrepancies between the models.

From Table 4, it is evident that the NDVI and the DI have relatively high loadings in both regions, corresponding to their significant spatial variations observed in Fig. 5. In contrast, the WET and the NDSI have lower loadings, consistent with their relatively minor spatial variations. Additionally, the LST exhibits greater variation in Beijing, leading to higher loadings. In contrast, Guangzhou’s LST shows less variation, resulting in lower loadings. This supports the reasonable assumption that an indicator’s loading magnitude depends on its variation extent within a region.

Table 4 Loadings of indicators for DRSEI and RSEI in Beijing and Guangzhou.

Index	DRSEI (Beijing)	RSEI (Beijing)	DRSEI (Guangzhou)	RSEI (Guangzhou)	
NDVI	0.68	0.72	0.67	0.98	
WET	0.13	0.14	0.08	0.12	
NDSI	−0.01	−0.01	0.00	0.00	
DI/LST	−0.72	−0.68	−0.74	−0.19	
Notes.

Note: The last indicator in DRSEI is DI, while the corresponding indicator in RSEI is LST.

Although DI and LST maintain strong correlations with EI, he accuracy of DRSEI and RSEI differs significantly in Guangzhou. Specifically, the RSEI tends to overestimate EQ while the DRSEI underestimates it. This occurs because LST’s low variability in Guangzhou reduces its RSEI loading. Consequently, LST’s negative impact on EQ diminishes, elevating RSEI scores. Conversely, DI’s high variability in Guangzhou increases its DRSEI loading. This amplification of negative influence on EQ leads to lower DRSEI scores. This further demonstrates that the spatial heterogeneity of indicators directly affects their weighting in the models, ultimately influencing EQ evaluation results.

Spatial and temporal changes in ecosystem quality of the city proper in Chongqing

Given the superior evaluation accuracy of the DRSEI across various districts in Chongqing metropolitan area, this index was selected to a analyze changes in ecosystem quality within the region. To visually represent the spatial distribution changes in EQ, the DRSEI values were classified into five ecological EQ levels (Poor, Fair, Moderate, Good, and Excellent) at intervals of 0.2. The classification results are shown in Fig. 6.

Figure 6 Spatial distribution of EQ levels.

As shown in Fig. 6, the ecological areas classified as ‘Moderate’ in 2002 largely transitioned to ‘Good’ by 2006. Between 2006 and 2010, regions with DRSEI values classified as ‘Moderate’ were primarily located in the western part of the study area, while a large portion in the eastern region improved to ‘Excellent’, indicating significant ecological enhancement in the east. However, from 2010 to 2014, the extent of ‘Excellent’ regions in the eastern part of the study area decreased substantially, with most of them downgraded to ‘Good,’ while the EQ in the central urban area further deteriorated to ‘Fair.’ This suggests that urban expansion may have exerted significant pressure on the ecological environment. Between 2014 and 2018, a large portion of the western area classified as ‘Good’ experienced slight ecological degradation, with most areas shifting from ‘Good’ to ‘Fair.’ From 2018 to 2022, the degradation trend persisted. The ‘Fair’ areas expanded westward, while the eastern region declined from ‘Good’ to ‘Moderate.’ These changes reflect an overall continuous decline in EQ.

To further quantify specific EQ changes in Chongqing’s city proper, the magnitude of changes was categorized into three levels (improved, basically stable, deteriorated) which were then expanded into five categories. The ‘improved’ category was further divided into ‘slightly improved’ and ‘markedly improved,’ while the deteriorated category was further divided into ‘mildly deteriorated’ and ‘markedly deteriorated.’ The specific classifications and their corresponding data were outlined in Table 5.

Table 5 Details of EQ grade changes.

Year	Level	Level area	Category	Category area	
02_06	Deteriorated	434.82	Mildly deteriorated
(–1, –2)	126.77	
Markedly deteriorated
(–3, –4)	308.05	
Basically stable	20229.27	Basically stable	20229.27	
Improved	12413.60	Slightly improved
(1,2)	12384.48	
Markedly improved
(3,4)	29.13	
06_10	Deteriorated	1695038.00	Mildly deteriorated
(–1, –2)	81.87	
Markedly deteriorated
(–3, –4)	1443.66	
Basically stable	27644.20	Basically stable	27644.20	
Improved	3872.83	Slightly improved
(1,2)	3770.80	
Markedly improved
(3,4)	102.02	
10_14	Deteriorated	9506.67	Mildly deteriorated
(–1, –2)	99.52	
Markedly deteriorated
(–3, –4)	9407.15	
Basically stable	23061.99	Basically stable	23061.99	
Improved	499.76	Slightly improved
(1,2)	465.02	
Markedly improved (3,4)	34.74	
14_18	Deteriorated	4449.26	Mildly deteriorated
(–1, –2)	51.78	
Markedly deteriorated
(–3, –4)	4397.49	
Basically stable	25347.93	Basically stable	25347.93	
Improved	3274.67	Slightly improved
(1,2)	3237.93	
Markedly improved
(3,4)	36.74	
18_22	Deteriorated	5068.94	Mildly deteriorated
(–1, –2)	54.61	
Markedly deteriorated
(–3, –4)	5014.32	
Basically stable	25170.56	Basically stable	25170.56	
Improved	2836.64	Slightly improved
(1,2)	2807.61	
Markedly improved
(3,4)	29.04	

Table 5 shows that between 2002 and 2006, the ecological environment exhibited an overall improvement trend. The area classified as ‘improved’ significantly exceeded the ‘deteriorated’ area, with ‘slightly improved’ being the dominant category. This indicates an enhancement in EQ during this period. However, between 2006 and 2014, the trend reversed. The ‘deteriorated’ area markedly outpaced the ‘improved’ area, with ‘markedly deteriorated’ becoming the predominant classification. This reflects a sharp ecological decline during this period. From 2014 to 2022, although the ‘deteriorated’ remained larger than the ‘improved’ area, the gap between the two narrowed significantly compared to the previous period. Ecological degradation continued to be driven by ‘markedly deteriorated,’ while recovery primarily consisted of ‘slightly improved.’ Overall, 2002–2006 was the only period of overall EQ improvement in the study area, while 2006–2022 exhibited a continuous degradation trend. Notably, the extent of ecological deterioration was generally significant, whereas ecological recovery was relatively slow and limited. This suggests that over the past two decades, the recovery capacity of the ecosystem has struggled to offset the negative impacts of human activities and environmental changes.

To further reveal the spatial migration patterns of EQ, changes were visualized through ArcGIS software. Specific pathways of ecological pattern adjustments were explored, as shown in Fig. 7.

Figure 7 Transition of the mean centers of the EQ grades.

Overall, except for the ‘Excellent’ EQ level, which remained stably distributed in the southeastern Nanchuan District, the mean centers of all other levels were concentrated near the central area of the study region. Notably, the ‘Poor’ EQ level showed the strongest spatial volatility. It fluctuated near the study area’s center, with transient westward expansion to Rongchang District and eastward shift to Banan District. In contrast, other EQ levels exhibited constrained spatial displacements, mainly oscillating around the central area.

Although the spatial distribution of EQ levels underwent varying degrees of change between 2002 and 2022, by 2022 the mean centers of most levels had reverted to positions proximate to their 2002 baselines. This shift suggested two concurrent trends: the ecological pattern fluctuated over two decades while also exhibiting signs of stabilization. This may indicate that urban development and ecological management have prevented sustained expansion/contraction trends. Instead, spatial evolution adapted to existing ecological patterns.

Discussion

In developing the RSEI model, Xu (2013a) used the ecological index (EI) as a reference for the initial selection of indicators. However, the RSEI did not include air pollution indicators from the EI (PRC, 2015) but instead used land surface temperature (LST) as an evaluation factor. This study refocused on EI’s air quality issue. We replaced the LST in the RSEI with the difference index (DI) that represents PM2.5 concentration (Feng, Feng & Feng, 2018). Compared to the RSEINew (Wan et al., 2021), ARSEI (Zhang et al., 2023), and AQRSEI (Wang & Ge, 2022) models mentioned in the introduction, as well as most studies that improve RSEI without validating their applicability across different geographic environments (Hu & Xu, 2018; Wang et al., 2020; Ye & Kuang, 2022), the proposed DRSEI offers the following advantages:

(1) The complexity of the DRSEI model does not increase. The DRSEI differs from models that add air quality indicators to expand the RSEI. By simply replacing LST with DI, it maintained model simplicity and kept the same number of indicators. This avoided introducing redundant information.

(2) The calculation process is more universal. DI had two advantages over the AOD and AQI used in ARSEI/AQRSEI: simpler computation and compatibility with various remote sensing data. This enhanced method generalizability. More importantly, the spatial resolution of DI is consistent with the other three indicators in the RSEI. This consistency ensures spatial compatibility among the indicators.

(3) The validation process is more comprehensive. This study not only evaluated their alignment with real ecological conditions through EI comparisons, but also systematically examined DRSEI-RSEI performance variations across cities. The investigation revealed differential applicability and limitations of these indices under varying environmental contexts.

For both RSEI and DRSEI, the loadings of individual indicators in the PCA did not entirely align with common expectations. This discrepancy warrants further investigation. Taking Guangzhou as a humid subtropical climate region, humidity is generally considered to have significant impacts on EQ. However, both RSEI and DRSEI showed relatively low loadings for the humidity-related WET indicator. The dryness indicator (NDSI) is partly based on built-up index. Given Guangzhou’s urbanization level, NDSI should have strong influence, but its model loadings were minimal. Its loading in both models was minimal to the point of being negligible. Based on the frequency distribution histograms of the indicators (Fig. 5), this phenomenon was attributed to the relatively low spatial variability of the humidity and dryness indicators within the study area. The contribution of a variable to PCA principal components depends on its overall dataset variance. If an indicator exhibits minimal spatial variation, its ability to explain variance in the principal components is also correspondingly low, resulting in a lower absolute loading value. This does not imply that humidity and dryness are unimportant for EQ. Rather, it indicates their limited capacity to differentiate overall EQ within this study area, resulting in reduced contributions to PCA. Therefore, the importance of an indicator should not be judged solely by its loading magnitude. Instead, a comprehensive analysis should be conducted based on its ecological significance and actual distribution characteristics. This finding suggests that when using PCA to construct ecological indices, it is essential to fully consider the spatial variability of each indicator to ensure the model accurately reflects EQ. For indicators with low spatial variability but high ecological significance, their impact should be carefully assessed during modeling to avoid overlooking their actual role due to low loadings. This approach improves the scientific validity and applicability of the model.

Moreover, the calculation of DRSEI may be affected by spatial resolution. Lower resolution can lead to the loss of details, impacting calculation accuracy (Xu et al., 2019). Since the loadings of DRSEI/RSEI indicators are influenced by spatial variability, if a certain indicator exhibits low spatial variation and the image resolution is also low, subtle changes may be overlooked, thereby affecting ecological quality assessment. However, this impact varies depending on the study scale. This study employs Landsat 30m imagery, demonstrating its reliability in municipal- and provincial-level ecological evaluations. If future studies adopt lower-resolution data (e.g., MODIS 500 m or 1 km), it may be necessary to expand the study scale to ensure effective capture of spatial variations in each indicator. Additionally, previous studies (Xu et al., 2018; Xu et al., 2019) have shown that mixed modeling of the four component indices at different scales can lead to information loss in the RSEI model. Therefore, the influence of spatial scale should not be overlooked in RSEI/DRSEI research. Temporal variations may also impact DRSEI calculations. Although this study does not specifically analyze this factor, existing research indicates that the accuracy of RSEI is significantly affected by vegetation seasonality (Miao et al., 2024; Zhang et al., 2021), with summer being the optimal season for RSEI construction (Huang et al., 2024). Furthermore, Zheng’s study suggests that nearly all RSEI-based studies reveal regional LSES changes by calculating the mean RSEI over multiple time periods. However, as RSEI is inherently a normalized relative index, whether these manually normalized relative values can be compared at an absolute level remains to be further verified (Zheng et al., 2022). Therefore, future research could further explore the impact of spatial and temporal resolution on DRSEI calculations and select appropriate data based on application needs.

Although the improvements of the DRSEI over the RSEI in this study were significant, some limitations remain:

(1) The study area experienced frequent cloud cover, and the use of cloud-removal algorithms introduced additional errors, affecting the accuracy of EQ assessments for the target years.

(2) While the DI reflected regional PM2.5 concentration to some extent, its specific mechanism and applicability require further investigation.

(3) This study used the EI as the reference standard for EQ assessment to validate the effectiveness of the DRSEI. However, the EI is not the only method for evaluating EQ. Other widely used models, such as the PSR model (OECD, 1993) and the ecosystem quality index (EQI) (PRC, 2021), emphasize different aspects of ecosystem evaluation and may provide perspectives distinct from EI. Future studies should compare the DRSEI with these models. This would assess its applicability across frameworks and better reveal its strengths/weaknesses.

(4) The DRSEI in this study was applied to only a limited geographic area, leaving the evaluation of the applicability of DRSEI and RSEI insufficiently comprehensive. Future studies can further expand the scope of research to enhance the understanding of the applicability of this index.

Conclusions

The results of this study showed that the DRSEI matches the RSEI in EQ assessment. Moreover, it exhibited higher evaluation accuracy in certain regions. The key findings of this study are as follows:

1. Higher evaluation accuracy of the DRSEI: In the main urban area of Chongqing, the DRSEI outperformed the RSEI in terms of overall fitting accuracy and spatial consistency of EQ assessment. Compared to RSEI, DRSEI demonstrated a smaller standard deviation gap relative to EI, higher correlation with EI, and lower RMSE when compared to EI’s baseline. This suggested that, within the study area, the DRSEI could more accurately reflect the regional EQ.

2. Differences in applicable regions: The DRSEI showed significantly higher accuracy than the RSEI in highly urbanized areas. Conversely, the RSEI performed better in less urbanized areas like suburban regions. The DRSEI demonstrated greater accuracy when applied on larger spatial scales. Specifically, it provided more precise EQ evaluations at provincial levels than at district or county levels. Therefore, it is necessary to appropriately select ecological indices at different research scales to ensure the accuracy of the evaluation results.

3. Impact of indicator spatial distribution on loadings: The study revealed that the spatial variation of ecological factors affected their PCA loadings. This in turn influenced the model’s EQ assessment. For example, in Guangzhou, the small LST variation caused lower loadings in the RSEI. This reduced its contribution to principal components and may lead to EQ overestimation. Therefore, when constructing ecological index models, prioritizing those with balanced indicator loading distributions is crucial. This helps reduce excessive influence from individual indicators on final evaluations.

When choosing between the RSEI and the DRSEI for EQ assessment, it is essential to consider both the characteristics of the study area and the applicability of the indices. If the study area is highly urbanized, the DRSEI may be the better choice, whereas in less urbanized regions, the RSEI may be more suitable. Additionally, the findings highlighted the importance of considering principal component loading distributions when constructing ecological index models. This consideration helps prevent excessive influence from individual indicators on final evaluation results. Future research on the DRSEI should further refine its construction methodology to minimize the extreme impact of specific indicators on model calculations. Additionally, future research could assess the stability and applicability of DRSEI in dynamic ecological quality monitoring by employing datasets with varying spatial resolutions and temporal scales.

Additional Information and Declarations

Competing Interests

Author Contributions

Data Availability

The authors declare there are no competing interests.

Guolin Zhang conceived and designed the experiments, performed the experiments, analyzed the data, prepared figures and/or tables, authored or reviewed drafts of the article, and approved the final draft.

Honghai Kuang conceived and designed the experiments, authored or reviewed drafts of the article, and approved the final draft.

The following information was supplied regarding data availability:

The remote sensing image data is available at Zenodo: Zhang, G. (2025). Raw remote sensing images used for DRSEI research. [Data set]. Zenodo. https://doi.org/10.5281/zenodo.14972174.

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
