# Peer review of "Urban ecosystem quality assessment based on the improved remote sensing ecological index"

_PeerJ, doi:10.7717/peerj.19297_

## Round 0.1 · original submission · Major Revisions

The article should strengthen either its methodological focus or the actual result of the study. At present, the methodological novelty is highly questionable (this was also noted by the reviewers), and the actual results are of interest only to the local community of ecologists, but not to the global community of scientists. I hope that the reviewers' comments will help you improve this manuscript.

Reviewer 1 ·

Basic reporting

Review of the manuscript entitled Urban Ecosystem quality Assessment Based on the Improved Remote Sensing Ecological Index, the authors introduced an interesting approach to evaluating ecosystem quality by improving the remote sensing ecological index (RSEI) and introducing the difference index (DI) to create the DRSEI.

Experimental design

The concept of modifying the Remote Sensing Ecological Index (RSEI) by replacing the Land Surface Temperature (LST) with the Difference Index (DI) to create the DRSEI is not entirely new. Previous studies have already attempted to enhance the RSEI by incorporating various air quality indicators. For instance, the RSEINew and AQRSEI mentioned in the introduction have similar aims. The manuscript does not clearly demonstrate how the DRSEI significantly advances the field beyond these existing approaches.

Validity of the findings

The study compares the DRSEI and RSEI with the Ecosystem Status Index (EI) in Chongqing, Beijing, and Guangzhou, but the validation process appears limited. The use of the EI as a reference standard has limitations, as it requires complex data acquisition and calculations and is restricted to county-level or administrative boundaries.

Additional comments

The spatial and temporal analysis of the ecosystem quality changes in Chongqing's main urban area is somewhat superficial. The manuscript presents box plots, Sankey diagrams, and spatial distribution maps, but the interpretation of these results is limited.

Reviewer 2 ·

Basic reporting

Manuscript ID; PeerJ_109169
Title: Urban Ecosystem quality Assessment Based on the Improved Remote Sensing Ecological Index
Although the topic is of interest to the scientific community, this paper should be improved before being considered for publication in any academic journal. Authors should reconsider the main objective of the paper according to its content. They should try to synthesize and emphasize the main findings of the review contents and avoid long sentences. Additionally, the paper's structure needs improvement for better readability and coherence. In addition, the conclusion is not well-written and fails to summarize the findings and highlight their significance effectively.

Experimental design

Methodology; The study uses only six years of data (2002, 2006, 2010, 2014, 2018, and 2022). This limited temporal resolution might not fully capture long-term trends or short-term fluctuations in RSEI and temperature.

While visual analysis is mentioned for cloud removal, no quantitative validation (e.g., accuracy metrics or comparisons with independent datasets) is reported for the key outputs like RSEI or surface temperature.

Validity of the findings

It's a nice try; But, the authors should go for the English edition. Some of the discussions are repetitive. Try to solve this problem. Otherwise, it is boring to read the manuscript.

Additional comments

Abstract: The authors should revise the abstract; it is too general. Moreover, it could be further developed, as the article has a lot of interesting data. An informative and representative conclusion should be added to the abstract.

Keywords: It is crucial to revise the keywords, ensuring they are spelled correctly and avoid general, abbreviation, and plural terms and multiple concepts (avoid, for example, 'and', 'of'). This will help to maintain the precision and clarity of the manuscript.
e.g. Remote sensing ecological index (RSEI), Ecosystem status index (EI), Ecosystem quality (EQ), Google earth engine (GEE)
(redundancy, no need abbreviation)

You must provide all the figures in high resolution and make the labels and legends more legible.

Conclusion: The findings could be further developed; the article contains a lot of interesting data.

·

Basic reporting

None

Experimental design

None

Validity of the findings

None

Additional comments

This paper presents a notable advancement with the DRSEI, incorporating air quality indicators into ecological evaluation and offering new opportunities for remote sensing-based assessments. While the study effectively demonstrates the model’s utility in certain urban contexts, it lacks comprehensive validation across diverse environments, which limits the generalizability of its findings. Additionally, the reliance on principal component analysis raises questions about the true ecological relevance and importance of the selected indicators, which could benefit from further exploration and alternative methodologies.

However, the paper could benefit from addressing the following critical issues:

First of all, you need to enhance the quality and readability of all maps and figures to help readers understand all the details conveniently.
Explanation of the Fmask Technique: A detailed explanation of the Fmask algorithm, its working principles, and its role in cloud detection and masking is essential for readers unfamiliar with this process.
Equation Syntax: The use of # in equations appears incorrect and should be replaced with × for consistency and clarity.
Visual Analysis Support: Where the paper states that visual analysis of data before and after masking demonstrated effectiveness, including a figure would significantly enhance comprehension and support this claim.
Simplification of Image ID: Instead of using full Landsat dataset identifiers, abbreviations like LC05 and LC08 can streamline the text without loss of clarity.
Impact of Cloud Removal: The paper does not quantify how much cloud removal affects the final results, leaving a gap in understanding the method’s influence on the outcomes.
PM2.5 Dataset Clarifications: The description of the PM2.5 dataset is insufficient and raises questions. Are the data sourced from fieldwork, satellite products, or open data portals? If from open data, the specific source should be clearly stated for transparency and reproducibility.

·

Basic reporting

There is a lack of literature in the Discussion, which is a must for readers to follow and assess your results and critically understand the content of the paper.

English was written well, but the readability was very poor, and the logic between sentences needed to be strengthened.

The figures need to increase their diversities.

Experimental design

1. The introduction to the previous work about EQ assessment still needs to be more inclusive.
2. In the first paragraph, it is only mentioned that EQ was assessed using EI and EQI, which lacks depth when serving as a part of the research review. The literature review was done in a very superficial way and lacked deeper depth.
3. You mentioned in line 69 that when Zhang et al. tested the Resines, the comparison result had some uncertainties. But you said that you applied the DI proposed by Feng et al. (2018). How did you consider the uncertainties in their product? And
4. What is the innovation of this study if only applying the index proposed by someone else? At least you should mention why replacing the LST with DI stands for its mechanism.
5. You only compared the DRSEI with EI, other than the comprehensive indices. It means that your analysis has limitations and requires more reliability. Also, why do you particularly select three cities?
6. Whether air pollution correlates highly with the DI, and whether adding DI brings new meanings to EQ, requires discussion.

Validity of the findings

1. How did you consider the consistency between the Landsat 5 and Landsat 8 images, and how these differences would affect your results?
2. The performance of DRSEI was analyzed through correlation calculation, however, it must have problems: such as the measurement of the effectiveness on the percentile that the novel index improved than the older version; I suggest the calculation of the sensitivity of DRSEI to different indices. And test the performance of DRSEI using multi-dimensional metrics, not only the correlation analysis.
3. Line 268-271 I don't quite understand this, what is the real purpose of DRSEI? You mentioned in the introduction about EQ, but here you mentioned that it can not reflect correctly EQ. Overall, the data you used was quite limited, with only 6 scenes.
4. The discussion part didn't use any literature to support the findings from your results. The future direction or the comprehensive suggestions for the readers to consider between RSEI and DRSEI are very unclear.
5. The conclusion part didn't highlight the key findings of this research work.

---

## Round 0.2 · Minor Revisions

Dear authors, I ask you to correct the minor comments in the manuscript and send it for approval for publication.

Reviewer 1 ·

Basic reporting

no comment

Experimental design

no comment

Validity of the findings

no comment

Additional comments

no comment

Reviewer 2 ·

Basic reporting

This revised version is suitable for publication.

Experimental design

-

Validity of the findings

-

Additional comments

-

Reviewer 5 ·

Basic reporting

The manuscript is well-structured and written in clear, professional English. The technical terminology is used appropriately, and the explanations are generally precise. While the language is fluent, minor refinements in phrasing—particularly in the methodology and results sections—could improve readability and flow. The introduction provides a solid foundation for the study, effectively outlining the challenges associated with urban ecosystem quality assessment and the limitations of traditional Remote Sensing Ecological Index (RSEI). The literature review includes relevant studies, establishing a clear context for the research. However, incorporating references to similar methodologies applied in diverse geographical settings could enhance the discussion. The paper follows a logical format, ensuring clarity from one section to the next. Figures and tables are well-prepared, clearly labeled, and informative. The authors have made the raw data available in accordance with PeerJ’s data-sharing requirements, facilitating reproducibility. The study is presented as a cohesive body of work, addressing a well-defined research question without unnecessary fragmentation.

Experimental design

The research falls within the scope of the journal and presents an innovative approach to urban ecosystem monitoring by enhancing RSEI with additional environmental parameters. The study effectively identifies an existing research gap—the limitations of traditional RSEI in capturing comprehensive urban environmental quality—and proposes an improved index (DRSEI) that incorporates air pollution data. The research question is well-articulated, demonstrating relevance to the field of urban environmental science. The methodology is rigorous, leveraging advanced remote sensing techniques, data processing within Google Earth Engine, and principal component analysis. These approaches are methodologically sound, but the description of certain steps—such as data preprocessing and the rationale behind the choice of specific statistical procedures—could be expanded for greater transparency. Ethical considerations are met, as the study relies solely on remote sensing data, eliminating direct human or ecological interventions.

Validity of the findings

The study’s findings are well-supported by robust data analysis. The research does not attempt to claim novelty for its own sake but instead builds upon existing methodologies to provide meaningful advancements in urban ecosystem assessment. The comparative analysis of RSEI and the newly developed DRSEI is well-executed, demonstrating the latter’s superiority, particularly in urbanized regions where traditional indices may fall short. The statistical analysis is comprehensive, confirming the improved performance of DRSEI. However, a more explicit discussion of potential methodological constraints—such as spatial resolution limitations or the impact of temporal variations—would further strengthen the study’s credibility. The conclusions align well with the research objectives, effectively demonstrating that DRSEI enhances urban ecosystem monitoring. The availability of all necessary data ensures that findings can be independently validated.

Additional comments

This study represents a valuable contribution to urban ecosystem research by refining an established assessment index to better capture environmental complexity. The methodological approach is robust, and the findings provide clear evidence of the benefits of incorporating air pollution data into ecosystem quality assessment. A deeper discussion on the applicability of this approach across different environmental and urban contexts would add further value. Some methodological details could be elaborated for better reproducibility, and minor refinements in phrasing would improve the manuscript’s clarity. Figures and tables are appropriately structured, but slight enhancements in visualization (e.g., improved contrast in maps or graphs) could aid readability. Overall, the study is well-conceived, rigorously conducted, and makes a meaningful contribution to environmental remote sensing.

---

## Round 0.3 · accepted · Accept

Dear Dr. Zhang and Dr. Kuang, I am pleased to inform you that your article has been accepted for publication.

Reviewer 5 ·

Basic reporting

The authors have revised the wording in the methodology and results sections, with specific changes detailed in the annotated revised version.The introduction has been supplemented by a discussion of similar advanced RSEI methods applied in different geographical environments to highlight the limitations of such studies in different contexts. This addition really lays the groundwork for testing the applicability of the DRSEI in different geographical environments. The Discussion section also discusses the application of the DRSEI in different regions, which is consistent with the introduction. These changes have significantly improved the quality of the manuscript.

Experimental design

The authors in the updated version of the manuscript have strengthened the rationale for data processing and the choice of specific statistical methods. An explanation of how to fill in the data removed from the clouds using images for the same month of neighbouring years was added. The authors provided a rationale for the use of the Taylor plot and its suitability for presenting the standard deviation, correlation coefficient and standard error. The corrections made have significantly improved the quality of the manuscript

Validity of the findings

The authors included an analysis of whether spatial resolution affects the DRSEI, and conclusions were drawn based on existing research. The authors also included a discussion of whether temporal variations affect DRSEI calculations. This study did not specifically analyse this aspect, so this question remains open. However, a citation was added to the manuscript and a conclusion was drawn to provide readers with an overview of the progress of research in this area.
I agree with the improvements made to the article.

Additional comments

The authors have done a great job of significantly improving the quality of the article. All the reviewer's recommendations have been taken into account. I recommend the article for publication.